# Organ-specific genome diversity of replication-competent SARS-CoV-2

Jolien Van Cleemput [1✉], Willem van Snippenberg [1,17], Laurens Lambrechts [1,2,17], Amélie Dendooven[3,4,5], Valentino D'Onofrio [6,7], Liesbeth Couck [8], Wim Trypsteen [1], Jan Vanrusselt[9], Sebastiaan Theuns[10,11], Nick Vereecke [10,11], Thierry P. P. van den Bosch[12], Martin Lammens[4,5], Ann Driessen[4,5], Ruth Achten[4,5,13], Ken R. Bracke [14], Wim Van den Broeck[8], Jan Von der Thüsen [12], Hans Nauwynck[11], Jo Van Dorpe [3], Sarah Gerlo[1,15], Piet Maes [16], Janneke Cox[6,7] & Linos Vandekerckhove [1✉]

Severe acute respiratory syndrome coronavirus 2 (SARS-CoV-2) infection is not always confined to the respiratory system, as it impacts people on a broad clinical spectrum from asymptomatic to severe systemic manifestations resulting in death. Further, accumulation of intra-host single nucleotide variants during prolonged SARS-CoV-2 infection may lead to emergence of variants of concern (VOCs). Still, information on virus infectivity and intra-host evolution across organs is sparse. We report a detailed virological analysis of thirteen postmortem coronavirus disease 2019 (COVID-19) cases that provides proof of viremia and presence of replication-competent SARS-CoV-2 in extrapulmonary organs of immunocompromised patients, including heart, kidney, liver, and spleen (NCT04366882). In parallel, we identify organ-specific SARS-CoV-2 genome diversity and mutations of concern N501Y, T1027I, and Y453F, while the patient had died long before reported emergence of VOCs. These mutations appear in multiple organs and replicate in Vero E6 cells, highlighting their infectivity. Finally, we show two stages of fatal disease evolution based on disease duration and viral loads in lungs and plasma. Our results provide insights about the pathogenesis and intra-host evolution of SARS-CoV-2 and show that COVID-19 treatment and hygiene measures need to be tailored to specific needs of immunocompromised patients, even when respiratory symptoms cease.

[1] HIV Cure Research Center, Department of Internal Medicine and Pediatrics, Ghent University Hospital, Ghent University, Ghent, Belgium. [2] BioBix, Department of Data Analysis and Mathematical Modelling, Faculty of Bioscience Engineering, Ghent University, Ghent, Belgium. [3] Department of Pathology, Ghent University Hospital, Ghent University, Ghent, Belgium. [4] Department of Pathology, Antwerp University Hospital, Edegem, Belgium. [5] Faculty of Medicine and Health Sciences, University of Antwerp, Wilrijk, Belgium. [6] Faculty of Medicine and Life Sciences, Hasselt University, Hasselt, Belgium. [7] Department of Infectious Diseases and Immunity, Jessa Hospital, Hasselt, Belgium. [8] Department of Morphology, Faculty of Veterinary Medicine, Ghent University, Merelbeke, Belgium. [9] Department of Radiology, Jessa hospital, Hasselt, Belgium. [10] PathoSense BV, Lier, Belgium. [11] Department of Virology, Parasitology and Immunology, Faculty of Veterinary Medicine, Ghent University, Merelbeke, Belgium. [12] Department of Pathology, Erasmus MC, Rotterdam, The Netherlands. [13] Department of Pathology, Jessa hospital, Hasselt, Belgium. [14] Laboratory for Translational Research in Obstructive Pulmonary Diseases, Department of Respiratory Medicine, Ghent University Hospital, Ghent University, Ghent, Belgium. [15] Department of Biomolecular Medicine, Ghent University, Ghent, Belgium. [16] Laboratory of Clinical and Epidemiological Virology, Department of Microbiology, Immunology and Transplantation, Rega Institute, KU Leuven, Leuven, Belgium. [17] These authors contributed equally: Willem van Snippenberg, Laurens Lambrechts. ✉email: jolien.vancleemput@ugent.be; linos.vandekerckhove@ugent.be

The coronavirus disease 2019 (COVID-19) pandemic, caused by severe acute respiratory syndrome coronavirus 2 (SARS-CoV-2), has now gripped the world for over a year and a half. The viral zoonotic origin, airborne transmission capacity, and genome plasticity have favored a fast and global human-to-human transmission of SARS-CoV-2 resulting in waves of epidemics[1–3]. People infected with SARS-CoV-2 may experience various symptoms ranging from mild respiratory illness and loss of smell and taste to severe respiratory and systemic manifestations resulting in death[4]. Besides, extensive and/or prolonged SARS-CoV-2 replication in the airways of immuno-compromised individuals accelerates viral intra-host evolution leading to the emergence and spread of new virus variants with higher transmission capacities (i.e., variants of concern [VOCs])[5–9].

Although the airways and lungs are considered as the "viral ground zero", SARS-CoV-2 is not always confined to the respiratory tract. Indeed, autopsy series have revealed several pathways to death in COVID-19 patients, including multi-organ failure. Besides, multiple studies have found traces of SARS-CoV-2 (i.e., viral RNA and proteins or virus-like particles) in various organs besides the lungs[10]. Still, current knowledge on complex virological parameters or intra-host evolution is limited to the respiratory system, while detailed information on viral presence in extrapulmonary compartments is missing[10–12].

Therefore, we have performed a detailed virological analysis of minimal invasive autopsy material from 13 COVID-19 patients collected at the Jessa Hospital in Hasselt, Belgium. Our results demonstrate viremia and dissemination of infectious SARS-CoV-2 to multiple extrapulmonary organs including the heart, kidney, liver, and spleen. Further, the study shows organ-specific SARS-CoV-2 genome diversity in an immunocompromised patient with long-term COVID-19. Moreover, we identify SARS-CoV-2 variants that have evolved hallmark mutations of current Alpha, Beta, and Gamma VOCs in multiple organs, while the patient had died long before the reported emergence of these variants.

## Results

The lungs and selected extrapulmonary organs (heart, kidney, liver, and spleen) of 13 deceased COVID-19 patients were biopsied postmortem under computed tomography (CT) guidance at the Jessa Hospital, Hasselt, Belgium during the period of April 15–June 30 2020. Additionally, plasma was collected from aortic blood and fractionated by ultracentrifugation. The mean age of this cohort was 77 (range 64–85) with five females and eight males. Patient clinical information are summarized in Table S1.

**Viral loads in the lungs and RNAemia are higher when patients succumb rapidly to infection**. Viral RNA was detected in the lungs and plasma of 9/13 patients (Table S1 and Fig. 1a). Cases were then stratified based on the duration of the disease (short <20 days; long >20 days). Patients who succumbed within 20 days following onset of symptoms generally had a higher viral RNA load in the lungs and plasma when compared to those that had lived longer ($P = 0.002$; $F = 12.589$; df = 23; Fig. 1a and Fig. S1). Our results support two phases of fatal disease evolution, including (i) short-lived disease with high viral loads in lungs and plasma, associated with a histological pattern of acute exudative alveolar damage in the lungs and (ii) long-lived disease with low (or undetectable) viral loads in lungs and plasma, associated with a chronic pattern of lung injury (Fig. 1a, b and Fig. S2). These results agree with the findings of another autopsy study[12]. Similarly, intra- and extracellular presence of SARS-CoV-2 nucleocapsid protein (NP) was more frequently identified in the lungs of cases with short-lived disease (four out of five), compared to those

with long-lived disease (one out of eight). Overall, bronchiolar epithelial cells and alveolar epithelial cells were the dominant cell type expressing SARS-CoV-2 NP, but we also identified viral NP in alveolar macrophages (Fig. 1b and Fig. S3), as described by others[10,12,13]. Patient 13, who was under rituximab treatment for B cell lymphoma at the time of infection, did not follow the trend and had an exceptionally high viral RNA load in the lungs ($10^{6.5}$ copies/40 ng RNA) and plasma ($10^{6.2}$ copies/mL) after 88 days of disease. SARS-CoV-2 NP was ubiquitously and abundantly found intracellularly and extracellularly in hyaline membranes in the latter patient's lungs, which showed a "remodeling pattern" with interstitial fibrosis and consolidation of airspace (Fig. 1b and Fig. S3).

**In immunocompromised patients replication-competent SARS-CoV-2 spreads systemically and disseminates to extrapulmonary organs**. The frequent detection of SARS-CoV-2 RNAemia (9/13) in this cohort indicates that systemic dissemination of viral components is quite common in severe COVID-19 cases, as described previously[14]. However, viremia (i.e., circulation of infectious virions) was found only in one case (patient 13). In this case, Vero E6 cells showed cytopathogenic effects upon inoculation with the plasma pellet and produced SARS-CoV-2 virions in the supernatant, as observed with light and transmission electron microscopy, respectively (Fig. 1c and S4a, c). These results do not exclude the possibility of viremia in other cases, as current isolation methods might not suffice to isolate virions when viral RNA loads in plasma are below a certain threshold.

Next, we identified two distinct types of disease progression based on viral RNA spread to extrapulmonary organs (i.e., heart, kidney, liver, and/or spleen), with intra-organ spread only occurring in 3 out of 13 cases (Fig. 1d). Digital droplet PCR was run on all biopsies of these three patients to quantify absolute SARS-CoV-2 copy numbers and confirmed the spread of viral RNA to multiple organs (Fig. S4b). Viral dissemination to multiple organs was strongly associated with profound immune suppression (chronic high-dose corticosteroid and/or rituximab treatment) at the time of or during infection (Fisher exact $P = 0.014$; Table S1). We hypothesize that inadequate immune responses during the early phase of SARS-CoV-2 infection resulted in enhanced viral replication and spread to extrapulmonary organs. Chronic high-dose corticosteroid treatment dampens viral-induced danger signals of the host immune response, resulting in the impaired release of critical antiviral components (e.g., interferons)[15,16]. Second, rituximab induces lysis and apoptosis of normal and malignant human B lymphocytes, essential for the production of virus-specific antibodies[17]. These findings point out the importance of patient management in severely immunocompromised COVID-19 patients.

Positive SARS-CoV-2 nucleocapsid (NP) staining was found in all organs of case 13, in renal and splenic tissue of case 07 and in splenic tissue of case 06. Viral NP was observed in cardiomyocytes and interstitial cells (heart), podocytes and tubular epithelial cells (kidney), hepatocytes, sinusoidal endothelial cells and Kupffer cells (liver), and myeloid cells (spleen) (Fig. 1e and Fig. S5). Viral RNA and proteins have been observed multiple times in myeloid cells, tubular cells, and podocytes in autopsy materials, but our results additionally provide evidence of hepatocytes and cardiomyocytes being in vivo SARS-CoV-2 targets[10,18,19]. Interestingly, SARS-CoV-2 NP was mainly detected in cell types expressing both the SARS-CoV-2 main receptor and co-receptor (i.e., angiotensin-converting enzyme type 2 [ACE2] and transmembrane serine protease 2 [TMPRSS2],

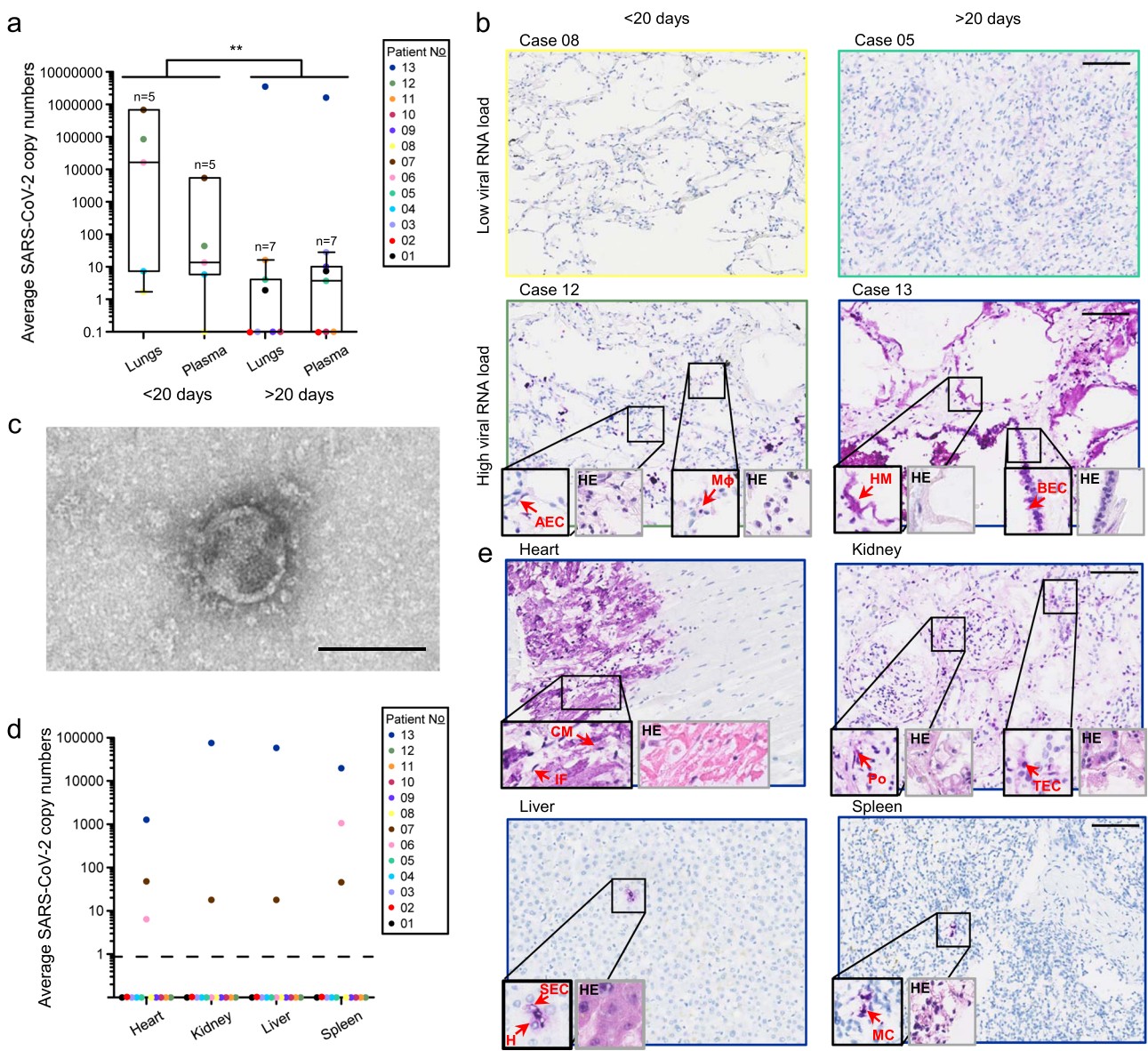

**Fig. 1 Detailed virological analysis of 13 postmortem COVID-19 cases. a** SARS-CoV-2 RNA loads, as measured with RT-qPCR, on a total of 40 ng RNA in the lungs or 1 mL of plasma of different cases (different colors) with short- or long-lived disease (<20 or >20 days after onset of symptoms, respectively). Central tendencies for SARS-CoV-2 copies are illustrated as a boxplot where the band indicates the median, the box indicates the first and third quartiles, and the whiskers indicate the minimum and maximum of all of the data. The significance of two-way ANOVA (with categorical variables duration and anatomical compartment) between the mean SARS-CoV-2 copy numbers of long and short duration is indicated on top (**$P = 0.002$; $F = 12.589$; df = 23). Source data are provided as a Source Data file. **b** SARS-CoV-2 nucleocapsid (NP) staining (in purple; large images and 2X-enlarged smaller images delineated in black) and corresponding hematoxylin-eosin (HE) staining (2X-enlarged smaller images delineated in gray) of paraffin-embedded sections of lung tissue of four different cases with low (<5 copies/40 ng RNA) or high (>5 copies/40 ng RNA) viral RNA loads and short- or long-lived disease. Arrows show specific SARS-CoV-2 NP-positive cells or hyaline membranes (HM). Picture linings correspond to the respective case colors shown in the legend of Fig. 1a. Lower magnification IHC and HE images are shown in Fig. S3. All scale bars represent 100 μm. AEC alveolar epithelial cell, BEC bronchiolar epithelial cell, HM hyaline membrane, Mϕ macrophage. **c** Transmission electron microscopic image of a SARS-CoV-2 particle from plasma-derived viral progeny on Vero E6 cells. Controls are shown in Fig. S4c. Scale bar indicates 100 nm. **d** SARS-CoV-2 RNA loads on a total of 40 ng RNA, as measured with RT-qPCR, in selected extrapulmonary organs. Source data are provided as a Source Data file. **e** SARS-CoV-2 NP staining (in purple; large images and 2X-enlarged smaller images delineated in black) and corresponding hematoxylin-eosin (HE) staining (2X-enlarged smaller images delineated in gray) of paraffin-embedded sections of extrapulmonary tissues of the case 13. Arrowheads indicate SARS-CoV-2 NP-positive cells. Picture delineations correspond to the respective case colors shown in the legend of Fig. 1d. Lower magnification IHC and HE images are shown in Fig. S5. CM cardiomyocyte, IF interstitial fibroblast, SEC sinusoidal endothelial cell, H hepatocyte, MC myeloid cell, Po podocyte, TEC tubular epithelial cell. Scale bars represent 100 μm.

respectively) across all organs examined (Fig. S6), confirming the in vivo relevance of ACE2 and TMPRSS2 in SARS-CoV-2 cell infection.

Further, we isolated virus from extrapulmonary organs that could replicate on Vero E6 cells. Infectious SARS-CoV-2 was isolated from the heart and kidney of case 07 and from all organs of case 13. These progeny viruses were subjected to full-length sequencing to confirm SARS-CoV-2 presence. The fact that we were unable to isolate infectious virus from SARS-CoV-2 RNA- and NP-positive splenic tissue in two out of three cases (case 06

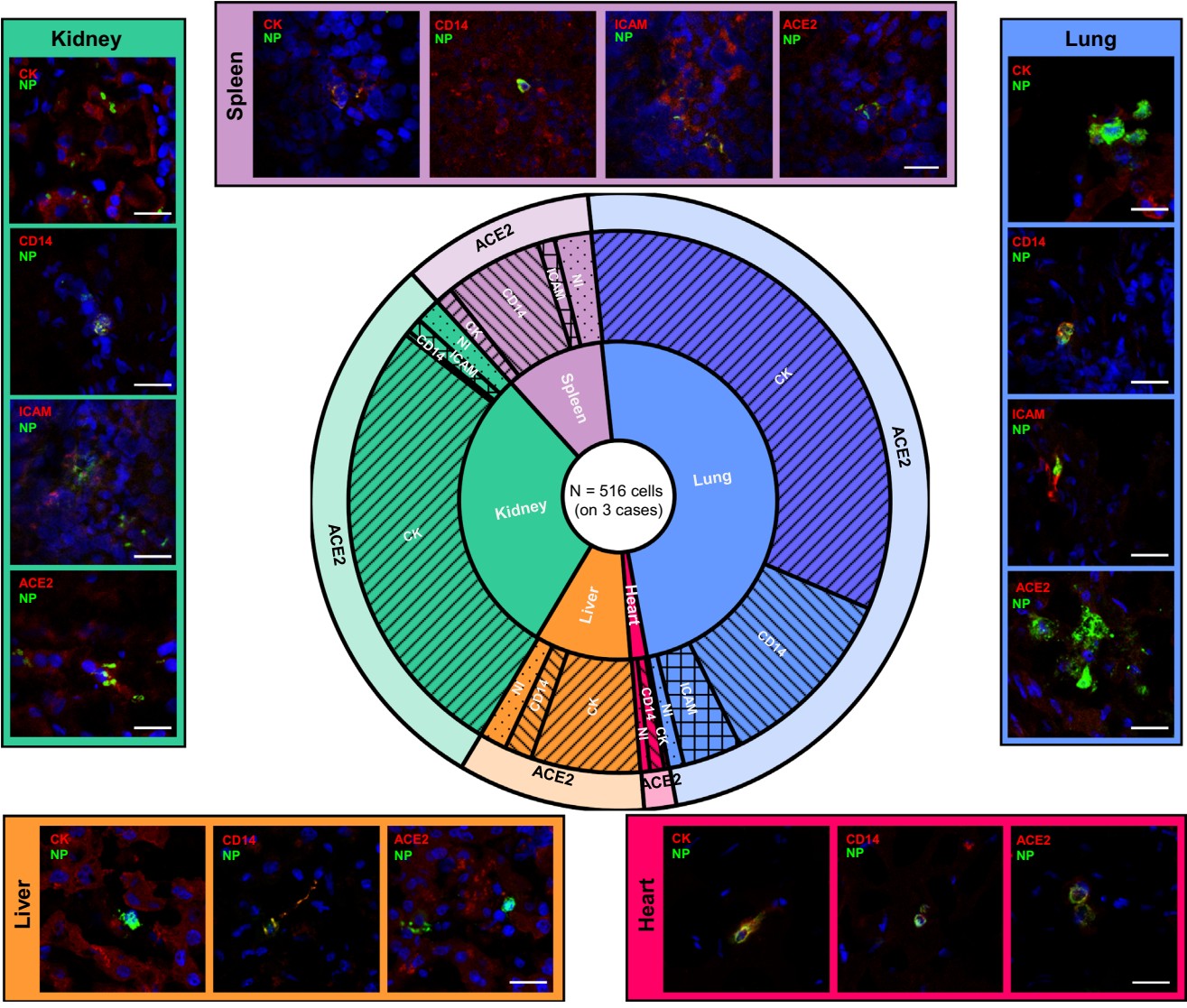

**Fig. 2 Delineation of SARS-CoV-2 NP-positive cells and total viral burden from postmortem biopsies, by tissue type.** A total of 516 SARS-CoV-2 nucleocapsid protein (NP)-positive cells were evaluated for marker expression (123 for ACE2, 133 for cytokeratin [CK], 136 for CD14, and 120 for ICAM) across different organs of three cases with viral dissemination (case 06, 07, and 13). The first level of the sunburst chart represents the distribution of SARS-CoV-2 NP-positive cells across different organs (blue = lung, red = heart, brown = liver, green = kidney, purple = spleen). The second level of the sunburst chart depicts the percentage of cell types positive for SARS-CoV-2 NP per organ (forward slash = cytokeratin [CK], backward slash = CD14, crosshatch = ICAM, dotted = not identified [NI]), and the third level shows co-localization data with ACE2 (long and short form). Source data are provided as a Source Data file. Representative confocal images are grouped per organ (different colors) at the outer edges. Scale bars represent 25 µm.

and 07) might indicate that the signal in these tissues derived from phagocytosed virions (and thus, viral RNA and proteins) in immune cells, rather than active viral replication in splenic cells. Viral loads in cardiac tissue of case 06 likely were too low for successful virus isolation. Of note, the presence of SARS-CoV-2 in extrapulmonary organs was rarely associated with pathological alterations in the respective organs, except for local cytolysis of cardiomyocytes in the heart, which was likely induced by viral replication. In addition, splenic lymphocyte depletion, which most likely was the result of the rituximab treatment, was apparent in case 13 (Fig. 1e and Fig. S5).

**SARS-CoV-2 nucleocapsid protein predominates in epithelial cells as well as cells from the myeloid lineage.** Although immunohistochemistry analysis can, to a limited extent, identify SARS-CoV-2 target cells, it does not allow marker co-localization at the cellular level. To identify SARS-CoV-2 tissue-specific target cell types in cases with intra-organ viral RNA dissemination (case 06, 07, and 13) we used double immunofluorescence staining and confocal microscopy. As shown in Fig. 2, the majority of SARS-CoV-2 NP-positive cells resided in the lungs (48.73% on a total of 516 positive cells), followed by the kidneys (29.69%), the spleen (10.98%), the liver (9.80%), and the heart (1.68%). SARS-CoV-2 NP was predominantly found in cytokeratin-positive (epithelial) cells in the lungs (67.90%), liver (66.63%), and kidneys (90.90%), while it was more commonly observed in CD14-positive (myeloid) cells in the heart (50.00%) and spleen (57.63%). ICAM-positive (endothelial) cells expressing SARS-CoV-2 NP occasionally were detected in the lungs (7.14%), kidney (3.00%), and spleen (10.02%). In general, SARS-CoV-2 NP was found only in ACE2-positive cells.

**Organ-specific SARS-CoV-2 evolution in an immunocompromised patient.** We hypothesized that SARS-CoV-2 replication in

multiple anatomical compartments would result in the emergence of new or specific variants in distinct organs, as described for other RNA viruses including poliovirus and HIV[20,21]. A recent study showed SARS-CoV-2 sequence diversity between respiratory and gastro-intestinal tract swabs from three COVID-19 patients[22]. In general, acute respiratory viral infections involve low intra-host diversity[23,24]. However, there is compelling evidence that SARS-CoV-2 evolution occurs in the respiratory tract of persistently infected immunocompromised hosts, reflecting prolonged virus replication and reduced selective immune pressure[5–9]. Therefore, we compared viral genome sequences from different organs in two patients with profound systemic and intra-organ viral spread (cases 07 and 13). Whole-genome sequencing was performed with Oxford Nanopore Technologies (ONT), shown to accurately detect SNVs and deletions in SARS-CoV-2 genomes[9].

For case 13, the phylogenetic analysis demonstrated that intra-host SARS-CoV-2 evolution occurred across multiple organs and confirmed that all SARS-CoV-2 genomes isolated from distinct anatomical compartments descended from a common ancestor derived from clade 20B (Fig. 3a). This is in contrast with the viral genomes derived from different anatomical compartments of case 07, which did not show evidence of viral evolution (Table S2). Despite the rapid spread of the infectious virus to extrapulmonary organs in the latter case, disease duration (3 days) was likely too short for virus evolution to occur. In contrast, case 13 only succumbed to SARS-CoV-2 infection after 88 days of disease, enabling prolonged viral replication resulting in high viral titers accompanied by viral evolution. The complete clinical history and disease course in case 13 is summarized in Fig. S7. In the latter case, consensus viral genomes retrieved from the kidneys had evolved the most mutations, followed by those found in lungs and heart, which in turn showed one additional mutation compared to those from spleen, liver, and plasma. A more detailed comparative analysis identified 50 (sub)consensus single nucleotide variations (SNVs) (18 synonymous and 32 non-synonymous mutations) and five deletions in viral genomes derived from different organs or plasma, as compared to clade 20B consensus genome (Fig. 3b, Fig. S8, Table 1, and Tables S3, 4). All mutations were verified with Illumina-based sequencing. These mutations were distributed in the 5′ and 3′ UTR and across seven out of ten protein-coding genes, including ORF1ab, S, E, ORF7a, ORF8, N, and ORF10. Three SNVs were fixed in all variants isolated from different compartments (frequencies >94%) and were, therefore, most likely present in the founder virus. In contrast, all other SNVs and deletions were detected at variable frequencies ranging between 1.11 and 98% depending on tissue origin, illustrating within-host organ-specific evolution of SARS-CoV-2.

Interestingly, several organs harbored viral populations distinct from all other compartments. For instance, four additional SNVs (T7247G, C7279T, and A8387G in ORF1a, and A27574T in ORF7a) were present at frequencies above 80% in the kidneys (Fig. 3b, Fig. S8, Table 1, and Table S3). In addition, six SNVs distributed across ORF1ab (A13433G, C16092T, T18024C, T18750C, and C18979T) and ORF10 (C29592T) were almost uniquely retrieved from kidneys. Six out of these ten SNVs were non-synonymous inducing amino acid substitutions in viral proteins including NSP3, NSP14, ORF7a, and ORF10. Still, viral infection capacity was not reduced by the majority of these SNVs (5/6), as these mutations were also identified in the viral progeny of Vero E6 cells inoculated with kidney-derived viruses (Table 1 and S3, in bold). NSP3, NSP14, and ORF7a are involved in viral protein processing, viral release, genome replication, and immune evasion, while the in vivo role of ORF10 is still under debate[25–28]. Variation in these proteins likely arose during extensive viral replication and spread in the kidneys, as evidenced by a large

number of SARS-CoV-2 NP-positive cells in the kidney (Fig. 2), and may have favored infection of the kidney following bottleneck events and viral adaptation to local environments.

Similarly, the viral population in the spleen was characterized by several unique SNVs that did not affect viral infectivity. For instance, we identified mutations in ORF1ab (C12513T [T4083M amino acid substitution in NSP8] and C14937T [no amino acid substitution in RNA-dependent RNA polymerase]), E (C26351T [A36V amino acid substitution in E protein]), and 3′ UTR (G29744A) with frequencies ranging between 33.61 and 64.31 % (Table 1 and S3). Specific alterations in NSP8 and E protein may favor viral infection or propagation in splenic tissue, as these viral proteins are involved in viral replication and budding[29].

Interestingly, in the kidney and liver, up to 40% of the viral genomes displayed the A23063T (N501Y amino acid substitution in S protein) alteration, a key mutation found in Alpha, Beta, and Gamma variants of concern (VOCs; Alpha 202012/01 [B.1.1.7], Beta GH/501Y.V2 [B.1.351], and Gamma GH/501Y.V3 [P1] lineages) that promotes viral binding, infectivity, and virulence[30–32]. In addition, this mutation is associated with adaptation to rodents[30]. The same SNV was also present in genomes derived from other organs, including the lungs, but at lower frequencies. Still, this mutation remained present in viruses propagated in Vero E6 cells from all tissues, highlighting the infection and transmission capacity of mutant N501Y viruses.

Similar to the N501Y mutation, we identified the C24642T (T1027I amino acid substitution) mutation in S, present in current strains of the Gamma GH/501Y.V3 [P1] lineage, at peaking concentrations of 50% in lungs and plasma, as well as in their viral offspring in Vero E6 cells[32]. In addition, a high SNV variability was detected in viral S genes derived from the lungs, which included mutation A22920T at a frequency of 52.52% leading to Y453F amino acid substitution in RBD of S protein. Remarkably, this mutation has been suggested to be the hallmark of the "mink variant". It is believed to increase viral binding to mink ACE2, and presumably also human ACE2[33]. However, since genomes with this SNV did not replicate in Vero E6 cells, binding and entry in African green monkey cells may be reduced. Our results suggest that viral evolution in the respiratory tract, but also in extrapulmonary organs of immunocompromised COVID-19 patients may prompt the emergence of more virulent and contagious SARS-CoV-2 variants with the capacity to infect other hosts. These findings are particularly interesting given the fact that our patient had died long before the reported emergence of VOCs, revealing convergent evolution of the latter mutations. Similar mutation events may still happen in the future, even when such patients are vaccinated. Indeed, vaccine effectiveness is less assured or even absent in immune-suppressed individuals, hence they remain highly susceptible to SARS-CoV-2[34]. Thus, these results highlight the utmost importance of hygienic and preventive measures to avoid viral spread from and to immune-suppressed patients.

Besides multiple SNVs, SARS-CoV-2 genotypes in renal, splenic, and hepatogenic tissue displayed a 91 bp deletion in ORF8 and/or deletions of varying size in the N-terminal tail of S protein of which some comprised the receptor-binding domain (RBD) (Fig. S9 and Table S4). However, none of these deletion mutations grew on Vero E6 cells, questioning their in vivo infectivity. The fact that the same deletion in ORF8 was present in multiple anatomical compartments, including plasma, indicates the systemic spread of the latter variant. In contrast, S deletion mutations differed among multiple organs and were absent in plasma. Still, how these deletion mutants accumulate simultaneously in multiple organs remains to be elucidated. In this context, similar variants with deletions in ORF8, but not S, have been detected in patients from different countries and have been

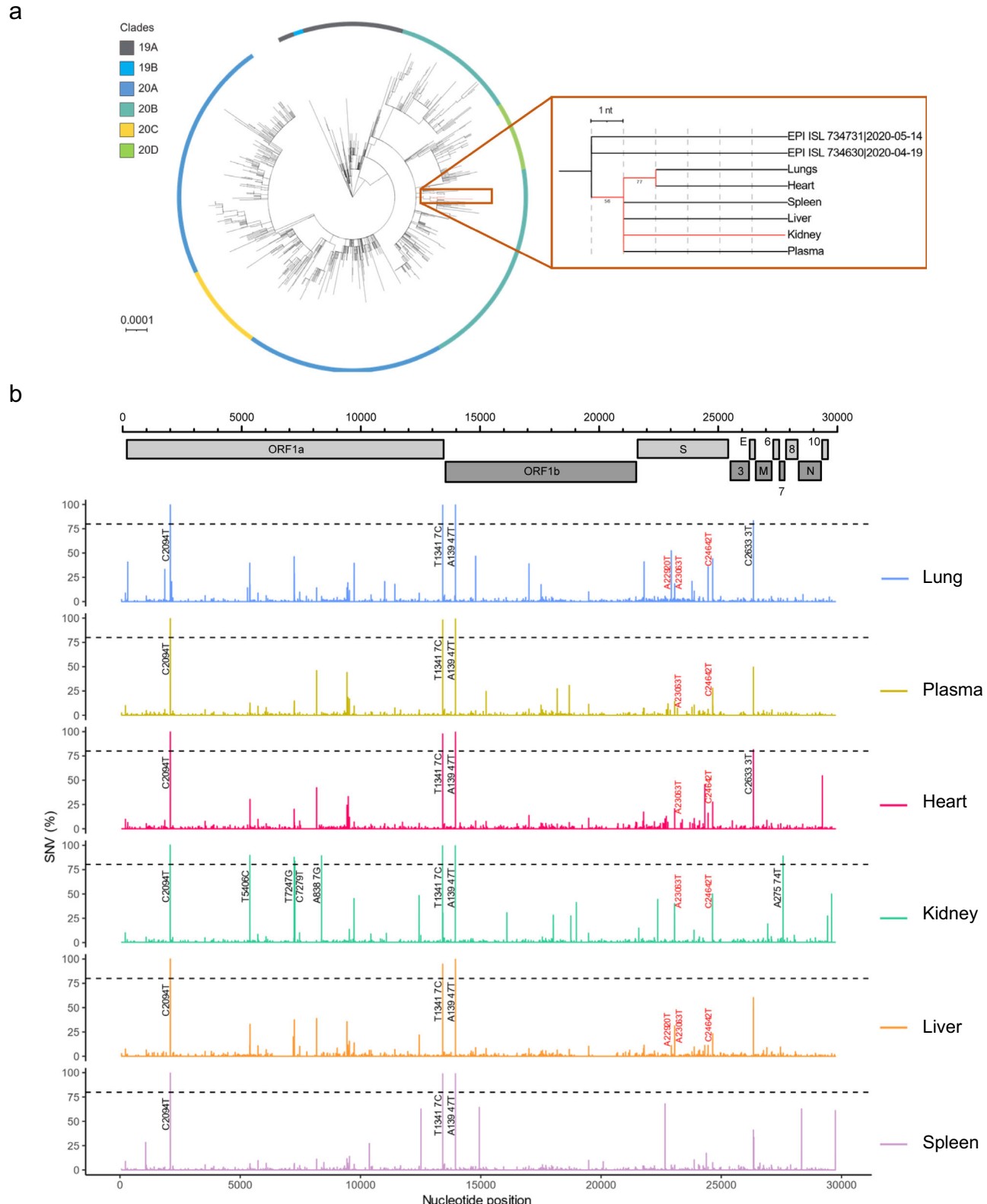

**Fig. 3 Tissue-specific SARS-CoV-2 evolution in an immune-suppressed individual with profound viral spread. a** Left: A circular maximum-likelihood phylogenetic tree rooted against the Wuhan-Hu-1 reference sequence, including SARS-CoV-2 consensus genomes from case 13 (in red) and public Belgian genomes from GISAID sampled between January 2020 and June 2020 (see also Table S6). The scale is proportional to the number of substitutions per site. Right: A detailed sub-tree highlighting case 13, displaying the underlying relation between the different anatomical compartments. Bootstrap values above 50 are shown. GISAID references are given in Table S6. **b** SARS-CoV-2 genome variations as compared to clade 20B consensus genome listed per anatomical compartment (different colors). Nucleotide positions and single nucleotide variation (SNV) frequencies are indicated on the X-axis and the Y-axis, respectively. SNVs with frequencies above 80% are annotated in black. SNVs with frequencies above 10% that are associated with variants of concern (VOCs) are annotated in red. A complete list of SNVs with allele frequencies per anatomical compartment can be found in Tables 1 and S3.

**Table 1 SARS-CoV-2 genome single nucleotide variants (SNVs) in postmortem tissues and plasma of case 13.**

| Gene[a] | Nt change | AA change | Frequency of detection per anatomical compartment (%)[b] | | | | | |
|---|---|---|---|---|---|---|---|---|
| | | | **Lungs** | **Plasma** | **Heart** | **Kidney** | **Liver** | **Spleen** |
| ORF1a | T322A | / | 40,81 | 0 | 6,57 | 0 | 0 | 0 |
| ORF1a | G1068A | NSP2:G268E | 0 | 0 | 0 | 0 | 0 | 28,5 |
| ORF1a | C1862T | NSP2:L533F | 33,28 | 5,9 | 8,78 | 0 | 0 | 3,11 |
| ORF1a | C2094T | NSP2:S610L | **99,56** | **99,55** | **99,62** | **100** | **100** | **99,62** |
| ORF1a | T2149C | / | 20,58 | 1,35 | <1 | <1 | <1 | <1 |
| ORF1a | T5395A | NSP3:F2328V | 39,67 | 5,95 | 4,12 | <1 | 11,9 | 4,03 |
| ORF1a | A5405G | NSP3:I1714V | 17,73 | 10,45 | 30,22 | 5,76 | 10,89 | 4,1 |
| ORF1a | T5406C | NSP3:I1714T | **10,63** | **12,5** | **28,95** | **89,21** | **32,82** | **6,63** |
| ORF1a | C7239T | NSP3:A2325V | 46,19 | 8,06 | 20,07 | <1 | 16,67 | 5,56 |
| ORF1a | T7247G | NSP3:F2328V | **29,24** | **14,66** | **6,78** | **87,36** | **37,5** | **6,77** |
| ORF1a | C7279T | / | **1,17** | 1,84 | <1 | **83,3** | <1 | <1 |
| ORF1a | C8175T | NSP3:A2637V | **14,31** | **45,94** | **42,28** | 1,85 | **38,91** | **11,11** |
| ORF1a | A8387G | NSP3:N2708D | **1,92** | **2,89** | <1 | **88,86** | 2,5 | 1,32 |
| ORF1a | C9438T | NSP4:T3058I | **14,4** | **44,24** | **24,27** | 1,11 | **35,56** | **11,64** |
| ORF1a | C9491T | NSP4:H3076Y | 2,84 | 1,28 | 33,29 | <1 | 5,65 | 4,05 |
| ORF1a | A9737G | NSP4:S3158G | **39,58** | **9,44** | **7,07** | **44,79** | **13,87** | **5,96** |
| ORF1a | C10369T | / | 1,54 | <1 | <1 | <1 | 6,04 | 27,3 |
| ORF1a | C11008T | / | 20,69 | 2,92 | 5,3 | <1 | 3,69 | 2,17 |
| ORF1a | C12439T | NSP8:P4058L | **8,71** | **5,35** | **5,76** | **48,05** | **21,86** | **4,07** |
| ORF1a | C12513T | NSP8:T4083M | <1 | <1 | <1 | 7,04 | <1 | **62,73** |
| ORF1a | T13417C | / | **99,25** | **98,09** | **97,72** | **99,15** | **94,8** | **98,89** |
| ORF1a | A13433G | NSP10:M4390G | **1,13** | **1,59** | <1 | **30,19** | 1,84 | <1 |
| ORF1b | A13947T | / | **99,5** | **99,17** | **99,46** | **99,26** | **99,64** | **98,96** |
| ORF1b | C14786T | RdRp:A440V | 46,74 | 4,66 | 8,13 | <1 | 8,85 | 3,66 |
| ORF1b | C14937T | / | <1 | <1 | <1 | <1 | <1 | **64,31** |
| ORF1b | C15222T | / | 1,84 | 24,59 | 7,67 | 1,24 | 7,97 | 2,96 |
| ORF1b | C16092T | / | <1 | <1 | <1 | 30,42 | <1 | <1 |
| ORF1b | C17004T | / | 39 | 6,46 | 13,84 | 0 | 7,85 | 4,12 |
| ORF1b | T18024C | / | <1 | <1 | <1 | 27,86 | <1 | <1 |
| ORF1b | A18179G | NSP14:K1571R | **<1** | **27,26** | **4,86** | <1 | **6,34** | **3,35** |
| ORF1b | T18678C | / | **<1** | **30,76** | **6,55** | <1 | **7,03** | **3,55** |
| ORF1b | T18750C | / | <1 | <1 | <1 | 27,13 | <1 | <1 |
| ORF1b | C18979T | NSP14:H1838Y | <1 | <1 | <1 | 40,81 | <1 | <1 |
| S | C21789T | S:T76I | 41,05 | 7,19 | <1 | <1 | 11,52 | 4,13 |
| S | G22363T | / | **4,65** | 2,14 | **<1** | **43,97** | 3,4 | 1,06 |
| S | G22661T | S:V367F | **5,62** | **<1** | **10,34** | **3,07** | **1,9** | **68,08** |
| S | A22920T | S:Y453F | 52,52 | <1 | <1 | <1 | 11,13 | 5,06 |
| S | A23063T | S:N501Y | **18,46** | **14,29** | **20,69** | **39,63** | **31,27** | **5,04** |
| S | G23782A | S:M740I | 20,87 | 8,26 | <1 | <1 | 4,55 | 1,72 |
| S | G24316T | S:E918D | **<1** | **3** | **45,65** | <1 | **11,33** | **1,94** |
| S | T24450C | S:V963A | 36,09 | 6,21 | 16,15 | <1 | 11,27 | 3,15 |
| S | C24642T | S:T1027I | **44,18** | **27,94** | **27,48** | **49,78** | **23,26** | **7,74** |
| E | C26333T | E:T30I | **83,41** | **49,57** | **81,39** | **2,25** | **60,29** | **41,17** |
| E | C26351T | E:A36V | <1 | **<1** | **<1** | <1 | <1 | **33,61** |
| ORF7a | A27574T | ORF7a:T61S | **2** | 2,1 | **<1** | **88,65** | 3,16 | <1 |
| N | A28336T | / | <1 | <1 | <1 | <1 | <1 | **62,93** |
| N | C29200T | / | <1 | <1 | 54,51 | <1 | <1 | <1 |
| N | A29424G | N:Q384R | <1 | <1 | <1 | 27,03 | <1 | <1 |
| ORF10 | C29592T | ORF10:T34M | **1,78** | 1,45 | **1,98** | **49,64** | 2,76 | <1 |
| 3′ UTR | G29744A | / | <1 | <1 | <1 | <1 | <1 | **61,12** |

Variants were mapped to clade 20B genome and called if case frequencies were higher than 20% in at least one anatomical compartment. SNV frequency is noted per organ.
[a]Gene names are italicized in the table.
[b]Bold numbers indicate that the latter variant was also identified in SARS-CoV-2 progeny on Vero E6 cells of respective tissue samples (frequencies >1%).

associated with milder infection[35–37]. Interestingly, porcine coronaviruses have been shown to shift tissue tropism or become less virulent due to deletions in S protein of similar sizes as deletions we observed in SARS-CoV-2 genomes from the liver[38]. Alternatively, it is possible that SARS-CoV-2 defective genomes, especially the ones with large deletions in *S*, might modulate viral replication or serve as immune decoys, thereby promoting viral persistence, as described for other RNA viruses[39]. We speculate that *S* deletion mutants may be involved in the viral occupation of

the kidney, spleen, and liver, but not in viral propagation in Vero E6 cells.

Finally, we attempted to link viral populations across different anatomical compartments based on haplotype frequencies. However, the tiling design and limited size of our amplicons impeded the phasing of mutations across the entire SARS-CoV-2 genome. The limited amount of linked SNVs did not provide any additional information about haplotype distribution across tissues.

## Discussion

Based on our comprehensive virological assessment of post-mortem COVID-19 cases and on previous autopsy series[10–12,40], we propose an adapted SARS-CoV-2 pathogenesis model in fatal COVID-19 disease. COVID-19 patients with severe disease initially suffer from extensive SARS-CoV-2 replication in the lungs, often accompanied with RNAemia. These patients may succumb rapidly to infection due to respiratory failure caused by acute exudative viral pneumonia with or without multi-organ failure resulting from the lack of oxygen and/or a detrimental virus-induced cytokine storm. Immunocompetent patients that mount an adequate antiviral response (innate and adaptive) may eventually clear the virus in the lungs and plasma and survive the initial phase of the disease. However, secondary (extra)pulmonary manifestations due to SARS-CoV-2 infection (e.g., airspace consolidation, bacterial superinfections, thrombosis, and sepsis) may still result in death later on. In contrast, the impaired antiviral response in immune-suppressed individuals paves the way for accelerated viral replication and multi-organ spread with organ-specific evolution. SARS-CoV-2 disseminates through the blood and infects ACE2- and TMPRSS2-expressing cells (e.g., epithelial cells such as tubular cells) at distinct locations. Here, virus replication is accompanied by further expansion of selected and unselected variants that facilitate colonization of the respective organ. Perhaps such organs may function as a viral reservoir, facilitating virus persistence and evolution. Given the emergence of specific mutants in distinct anatomical compartments, also found in currently circulating VOCs (lineages 202012/01, GH/501Y.V2, and GH/501Y.V3), highly transmissible SARS-CoV-2 variants may arise in such patients and potentially spread to other individuals. Extrapulmonary development of infectious and transmissible SARS-CoV-2 variants may grow in significance in immunocompromised individuals, as their immune response to vaccination is blunted. Hence, SARS-CoV-2 can still replicate and mutate extensively in those patients—also in extrapulmonary organs—giving rise to transmissible variants that may no longer be susceptible to vaccine-induced antibodies. Thus, (extra-pulmonary) SARS-CoV-2 replication and spread in and from these individuals will become more and more important to monitor in the future. Eventually, patients succumb to pathological alterations caused by extensive viral replication and cellular damage throughout the body, yet mainly in the lungs. These findings highlight the need for tailoring COVID-19 treatment strategies and isolation management to the phase of the disease and the patient's immune status.

## Methods

**Ethics statement**. All procedures performed in studies involving human subjects were in accordance with the ethical standards of the institutional and/or national research committee and with the 1964 Helsinki Declaration and its later amendments or comparable ethical standards. Documented approval was obtained from the Ethics Committees of Jessa hospital and Hasselt University (Clinicaltrials.gov identifier: NCT04366882). Oral consent for sample collection was obtained from the patients' legal representatives. Written consent could not be obtained due to visiting restrictions in the hospital during the pandemic, but written information was provided via registered mail to the patients' legal representative after oral informed consent.

In this manuscript, we report the results of secondary outcomes. These were "to describe the quantity of viral RNA in the different tissues and relate this to the clinical, radiological, and histopathological findings" and "to study in detail the disease mechanisms at the cellular level (including ACE2 receptor expression in relation to the quantity of viral RNA) in the different tissues". Additionally, several non-prespecified exploratory outcomes were added to the trial and reported here (TMPRSS2 expression, virus isolation, viral genome analyses), for which approval was obtained by the Ethical Committee of Jessa Hospital. The study protocol is available from the authors on reasonable request.

**Sample collection**. Minimally invasive autopsy (MIA) was performed on a total of 13 COVID-19 patients at Jessa hospital who succumbed to infection between April

15 and June 30, 2020. All patients were confirmed for SARS-CoV-2 infection through RT-qPCR analysis performed on nasopharyngeal swabs. Patient demographics and clinical information is summarized in Table S1. MIA was performed within 24 h of death[41]. Briefly, tru-cut biopsies were taken under computed tomography (CT)-guidance (14 G biopsy needle, C.R. Bard, Murray Hill, NJ, USA) from lungs, heart, liver, spleen, and kidneys. Tissue samples were either (i) snap-frozen and stored dry at −80 °C for cryosectioning and virus isolation, (ii) sub-merged in RNA-later before snap-freezing and stored at −80 °C for RNA and protein analyses, or (iii) fixed in 10% neutral buffered formalin for 72 h prior to embedment in paraffin for hematoxylin-eosin (HE) staining or immunohistochemistry (IHC). The quality of biopsies was confirmed through histological analysis. Blood was collected from the aorta and transferred to citrate or heparin tubes (Vacuette, Greiner Bio-One, Vilvoorde, Belgium). Plasma was collected from these tubes following centrifugation at $2500 \times g$ for 5 min at room temperature and stored at −80 °C.

**Tissue homogenization**. Tissue homogenates were prepared using bead mill technology by high-speed shaking (50 Hz) of tissues with 5 mm stainless steel beads for 5 min in a TissueLyser LT (Qiagen). Ten percent (w/v) solutions of single tissues were made in either (i) RLT buffer (#79216, Qiagen, Hilden, Germany) supplemented with 1% β-mercaptoethanol (#M6250, Sigma-Aldrich, St. Louis, MO, USA) for RNA analyses, (ii) RIPA buffer (#R0278, Sigma-Aldrich) supplemented with cOmplete™, EDTA-free Protease Inhibitor Cocktail (#11873580001, Roche, Basel, Switzerland) for protein analyses, or (iii) Dulbecco's modified Eagle medium (DMEM; #41965-039, ThermoFisher Scientific, Waltham, MA, USA) supplemented with 8% heat-inactivated fetal bovine serum (FBS; #SV30160.03, ThermoFisher Scientific), 0.075% sodium bicarbonate (#25080094, ThermoFisher Scientific), and 1 mM sodium pyruvate (#11360070, ThermoFisher Scientific) for virus isolation. Finally, tissue homogenates were clarified by centrifugation at $13,000 \times g$ for 10 min at 4 °C.

**Virion pelleting from plasma**. One milliliter of plasma (citrate tubes) was diluted in 10 mL of Dulbecco's phosphate-buffered saline (#14190144, ThermoFisher Scientific) and centrifuged at $45,000 \times g$ for 3 h at 4 °C using a Sorvall centrifuge (ThermoFisher Scientific) and A27 rotor. Pellets were resuspended in the same three buffers (200 µL) as described above for tissue samples.

**RNA extraction and cDNA synthesis**. RNA was extracted from clarified tissue homogenates and dissolved plasma pellets using the RNeasy® Plus Mini Kit (#74034, Qiagen) following the manufacturer's instructions. Prior to RNA extraction, genomic DNA was removed using the gDNA Eliminator spin columns. RNA quantities were measured using Qubit® RNA BR Assay Kits (#Q10211, ThermoFisher Scientific). A total of 1 µg RNA was reverse-transcribed to cDNA using the SuperScript III First-Strand Synthesis System (#18080051, ThermoFisher Scientific) following the manufacturer's instructions.

**qPCR**

*SARS-CoV-2*. Two microliters of cDNA was subjected in duplicate to quantitative PCR (qPCR) using a CDC qPCR probe assay (N1, Integrated DNA Technologies, Coralville, IA, USA, Table S5) with LightCycler® 480 Probes Master (#04707494001, Roche). A tenfold dilution series of 2019-nCoV Plasmid Control (Integrated DNA Technologies), corresponding to a range of 2 to $2 \times 10^5$ SARS-CoV-2 RNA copies, functioned as standard. qPCR was performed using a Light-Cycler® 480 Real-Time PCR System (Roche) with the following amplification conditions: preincubation at 95° for 10 min with 45 cycles of denaturation (30 s at 95 °C), annealing and elongation (30 s at 55 °C), followed by a final elongation for 5 min at 40 °C. LightCycler 480 v1.5.0.39 was used for data collection. Sample CT values were plotted against standard dilution values to determine exact SARS-CoV-2 RNA concentrations.

*ACE2 and TMPRSS2*. Two microliters of cDNA was subjected in duplicate to quantitative PCR (qPCR) using LightCycler® 480 SYBR Green I Master (#04707516001, Roche) and specific primers for ACE2[42] and TMPRSS2 (Table S5). Beta-actin (ACTB), glyceraldehyde 3-phosphate dehydrogenase (GAPDH), and YWHA were included as housekeeping genes for the normalization of gene expression[43]. No template sample was used as a negative control. The qPCR was performed on a LightCycler® 480 Real-Time PCR System (Roche) with the following amplification conditions: preincubation at 95° for 2 min with 45 cycles of denaturation (15 s at 95 °C), annealing (30 s at 60 °C), and extension (15 s at 72 °C), followed by a melting curve from 55 to 95 °C. LightCycler 480 v1.5.0.39 was used for data collection. Relative gene expression was calculated using qbase⁺ software v3.2 (Biogazelle, Zwijnaarde, Belgium).

**ddPCR**. Two microliters of cDNA was subjected in duplicate to droplet digital PCR (ddPCR) using a CDC qPCR probe assay (N1, Integrated DNA Technologies) with Master mix for probes (#1863010, Bio-Rad, Hercules, CA, USA). The digital PCR was performed using a QX100™ Droplet Digital™ PCR System (Bio-Rad) with the following amplification conditions: preincubation at 95° for 10 min with 40 cycles

of denaturation (30 s at 94 °C), annealing and elongation (30 s at 56 °C), followed by enzyme deactivation for 10 min at 98 °C. The ramp rate during the PCR was set at 2 °C per second. Samples were read-out using the QX100 Droplet Reader and analyzed with QuantaSoft software v1.6.6.0320 (Bio-Rad). Final copy numbers were determined using the ddpcRquant shiny tool in R with standard settings (https://ddpcrquant.ugent.be/)[44].

### Whole-genome sequencing and genome assembly

*Oxford Nanopore Technologies (ONT) sequencing.* Following RNA extraction as described above, cDNA was synthesized followed by multiplex PCR amplification using a modified version of the ARTIC V3 LoCost protocol with the Midnight primer set (1200 bp amplicons)[45–47]. An additional primer set (A6 with resulting in a 2500 bp amplicon) was used to PCR amplify part of the S gene[46]. The libraries were sequenced on a MinION using R9.4.1 flow-cells (Oxford Nanopore Technologies, Oxford, UK) and MinKnow software v21.02.1. The resulting fast5 reads were basecalled and demultiplexed using Guppy v4.2.2 in high accuracy mode. Genome assembly was performed using the ARTIC bioinformatics pipeline v1.1.3, which entails adapter trimming, mapping to the reference strain Wuhan-Hu-1 (MN908947 [https://www.ncbi.nlm.nih.gov/nuccore/MN908947.3]) and consensus calling with 20x minimum coverage (https://artic.network/ncov-2019/ncov2019-bioinformatics-sop.html)[48]. The mapping assembly of the viral genome was nearly complete (99.5%) for all samples with a minimum average of 200-fold read depth. SNVs on consensus level were identified via Nanopolish v0.13.2 and filtered by the ARTIC *artic_vcf_filter* tool v1.1.3 while SNVs at lower frequencies were identified using Varscan2 v2.4.3[49,50]. Additionally, to detect structural variants for each sample, an alignment to MN908947 was made via NGMLR v0.2.7 and subsequently used by Sniffles v1.0.11 to identify structural variants with a minimum size of 10 bp and ≥20 supporting reads[51]. Only SNVs with frequencies higher than 20% in at least one anatomical compartment were called, as Bull et al., 2020[9] previously showed that SARS-CoV-2 variants at read-count frequencies above 20% are highly accurate and genuine.

*Illumina sequencing.* Illumina sequencing was used to verify mutations and deletions identified with ONT-based sequencing. To do so, cleaned amplicon pools were prepped for Illumina sequencing using the Nextera XT DNA Library Preparation Kit (#FC-131-1096, Illumina) according to the manufacturer's instructions. The libraries were sequenced on a MiSeq Illumina platform via 2 × 150 nt paired-end sequencing with the 300 cycle v2 kit (#MS-102-2002, Illumina) according to the manufacturer's instructions. Data for Illumina sequencing was collected using BaseSpace Sequence Hub v6.8 software. A reference-based assembly for each individual sample was performed using the following steps: (i) a quality check of the data using FastQC v0.11.7 (http://www.bioinformatics.babraham.ac.uk/projects/fastqc). (ii) Removal of Illumina adapter sequences and quality-trimming of 5′ and 3′ terminal ends was performed with bbmap v37.99 (sourceforge.net/projects/bbmap/) followed by primer clipping of used amplicon primers using Cutadapt v3.4 (https://github.com/marcelm/cutadapt). (iii) Clipped and trimmed reads were mapped to the reference strain Wuhan-Hu-1 (MN908947) using bbmap v37.99 (sourceforge.net/projects/bbmap/). (iv) A final consensus sequence was generated using samtools v1.6 (http://www.htslib.org/). SNVs were identified using Varscan2 v2.4.3.

### Phylogenetic analysis

Public Belgian SARS-CoV-2 genomes with a high coverage were collected on March 12, 2021 from the GISAID database (https://www.gisaid.org/) with sampling dates ranging between January 2020 and June 2020 (Table S6). A genome alignment using MAFFT was constructed using the case 13 consensus genomes derived from different anatomical compartments, the Belgian GISAID genomes and the MN908947 reference[52]. Clades for each sequence were assigned via Nextclade (https://github.com/nextstrain/nextclade/). Using the nextstrain toolkit v6, a maximum-likelihood phylogenetic tree was constructed via IQ-Tree v1.6.9, using a GTR substitution model and performing 100 bootstraps with annotation and visualization done via iTol v6 (https://github.com/nextstrain/ncov/)[53,54].

### ELISA

Protein levels in clarified tissue homogenates were determined using the Pierce™ Detergent Compatible Bradford Assay Kit (#23246, ThermoFisher Scientific) according to the manufacturer's instructions. ACE2 and TMPRSS2 levels were quantified using the human ACE2 ELISA Kit (#ab235649; Abcam, Cambridge, United Kingdom) and human TMPRSS2 ELISA Kit (#NBP2-89171; Novus Biologicals, Centennial, CO, USA), respectively, according to the manufacturers' instructions. Finally, protein concentrations were normalized to total protein content (ng/mg total protein). Data were collected using a SpectraMax i3x and SoftMax Pro v7.1.

### Immunohistochemistry

Four-micrometer-thick formalin-fixed and paraffin-embedded sections were cut and subjected to immunohistochemistry. SARS-CoV-2 nucleocapsid protein (NP) was stained by automated IHC using the Discovery ULTRA platform (Ventana Medical Systems, Oro Valley, AZ, USA). Sections were deparaffinized prior to heat-induced antigen retrieval with CC1 (#950-500, Ventana Medical Systems) for 32 min. Next, slides were incubated with a rabbit polyclonal anti-SARS-CoV-2 nucleocapsid protein (NP) antibody (1:1000 dilution, #40143-T62; Sinobiological, Beijing, China) for 32 min at 37 °C. Detection was done with omnimap-anti-Rabbit HRP (undiluted, #760-4311, Ventana Medical Systems) for 16 min and visualized with Discovery Purple (undiluted, #760-229, Ventana Medical Systems) for 32 min. Incubation was followed by a hematoxylin II counterstain for 4 min and then a blue coloring reagent for 4 min according to the manufacturers' instructions (Ventana Medical Systems). ACE2 was immune-stained using monoclonal rabbit anti-human ACE2 antibodies that recognize the N-terminal domain of the long-form (i.e., the virus binding site; 0.5 μg/mL, #ab108252; Abcam)[55]. TMPRSS2 was stained using monoclonal rabbit anti-human TMPRSS2 (0.5 μg/mL, #ab109131; Abcam) following heat-mediated antigen retrieval in Tris/EDTA buffer pH 9. In the second step, horseradish peroxidase (HRP)-labeled poly anti-rabbit IgG antibodies (undiluted, #DPVR-55HPR; Immunologic, Duiven, The Netherlands) were added. Immunostaining was visualized using DAB+ (#K3467, Agilent, Santa Clara, CA, USA) and hematoxylin II was used for counterstaining. Positive controls were used on every slide. Rabbit monoclonal or polyclonal isotype antibodies functioned as negative controls (0.5 μg/mL, #ab172730 and #ab15348; Abcam). All slides were digitally scanned using the Hamamatsu NanoZoomer 2.0RS and analyzed using Hamamutsu NDP.view v2.9.25.

### Immunofluorescence staining and confocal microscopy analysis

Ten-micrometer-thick cryosections were double-stained using a monoclonal mouse anti-SARS-CoV-2 antibody (1:1000 dilution, #MBS569903; MyBioSource, San Diego, CA, USA) and one of the following antibodies: polyclonal rabbit anti-pan cytokeratin antibody (1:100 dilution, #ab9377; Abcam), monoclonal rabbit anti-CD14 antibody (1:100 dilution, #ab18332; Abcam), polyclonal rabbit anti-ACE2 (1:100 dilution, recognizing both short and long forms of ACE2; #PK-AB718-3217, PromoCell, Heidelberg, Germany), monoclonal rabbit anti-ICAM-1 (1:100 dilution, #ab109361; Abcam). Mouse monoclonal and rabbit monoclonal or polyclonal isotype antibodies (#ab18469, #ab172730 and #ab15348; Abcam) functioned as negative controls. In the secondary step, FITC-conjugated polyclonal goat anti-mouse antibodies (1:200 dilution #F2761, ThermoFisher Scientific) were combined with Texas Red-conjugated polyclonal donkey anti-rabbit antibodies (1:100 dilution, #ab6800, Abcam). In the tertiary step, FITC-conjugated polyclonal donkey anti-goat antibodies were added (1:200 dilution, #A16006, ThermoFisher Scientific). DAPI (#D9542, Sigma-Aldrich) was used to counterstain cell nuclei. Slides were mounted with Fluoroshield™ (#F6182, Sigma-Aldrich) and analyzed using a Leica (TCS SPE) confocal microscope and Leica Las X v3.7.2.22383 software.

### Transmission electron microscopy

Seven microliters of pelleted plasma resuspended in complete medium or plasma-derived viral progeny produced on Vero E6 cells was spotted onto copper hexagonal EM grids (#FCF200H-Cu-TB, Electron Microscopy Sciences, Hatfield, PA, USA). Complete medium only or plasma of a COVID-19-negative patient and a virus stock of porcine respiratory coronavirus (strain 20v17) functioned as negative and positive controls, respectively (Fig. S4c). Grids were washed one time in ultrapure water prior to negative staining with 1% uranyl acetate for 45 s. EM grids were observed using a JEOL JEM-1400 Plus transmission electron microscope and Radius v1.4 software.

### Virus isolation

Tissue homogenates and resuspended plasma pellets were added to 90% confluent Vero E6 cell (#C1008, ATCC, Manassas, VA, USA) monolayers (six-well format) and incubated on a plate rocker for 2 h at 35 °C. After two washes with an excess of sterile PBS, fresh DMEM supplemented with 2% heat-inactivated FBS and 2 mg/ml trypsin (#27250018, ThermoFisher Scientific) was added and plates were incubated in a humidified 5% CO2 atmosphere at 37 °C. Cells were daily monitored for cytopathogenic effect (CPE) for a maximum of 21 days. The cell medium was replaced every 7 days.

### Statistics and reproducibility

Every patient represents a single entity and individual values per case are shown in Fig. 1a, d and Figs. S1, S4b, S6a, b. RT-qPCR, ddPCR, and ELISA data were obtained in duplicate and averages are shown in Fig. 1a, d and Figs. S1, S4b, S6a, b. Histological analysis and stainings were performed on 1–5 sections per case and representative images are shown in Fig. 1b, e, Fig. 2 and Figs. S2, S3, S5, S6c. Virus isolations were performed once on tissue homogenates or plasma pellets of different samples per case (Fig. S4a). Electron microscopy was performed in duplicate on viral isolates from plasma and control samples (Fig. S4c). All PCRs to amplify whole genomes were performed once, except for the PCR using primer pair A6 targeting the partial S gene, which was performed twice and of which a representative image is shown in Fig. S9. Whole-genome sequencing was performed once on tissue samples and once on virus isolates derived from these tissues using both ONT and Illumina sequencing.

Significant differences were identified by a Student's *t*-test (two groups) or analysis of variance (ANOVA) followed by a Tukey post hoc test (multiple groups). If homoscedasticity of the variables was not met, as assessed by Levene's test, the data were log-transformed prior to *t*-tests or ANOVA. The normality of the residuals was verified using the Shapiro–Wilk test. If the variables remained heteroscedastic or normality was not met after log transformation, a Mann–Whitney's test (two groups) or Kruskal–Wallis test (multiple groups) was

performed. The significance of the associations were determined using Fisher's exact test. All analyses were conducted in IBM SPSS Statistics, version 27.0 (IBM Corp., Armonk, NY, USA).

**Reporting Summary**. Further information on research design is available in the Nature Research Reporting Summary linked to this article.

## Data availability

SARS-CoV-2 consensus genomes generated in this study have been deposited in the GISAID repository under the following accession codes: EPI_ISL_1404134, EPI_ISL_1404133, EPI_ISL_1404136, EPI_ISL_1404135, EPI_ISL_1404141, EPI_ISL_1404140, EPI_ISL_1404132, EPI_ISL_1404131, EPI_ISL_1404142, EPI_ISL_1404138, EPI_ISL_1404137, and EPI_ISL_1404139 (https://www.epicov.org/). The raw sequencing data generated in this study have been deposited in the SRA database under the following BioProject study ID: PRJNA724859. All other data generated or analyzed during this study are included in this paper and its Supplementary Information files. Data underlying Main and Supplementary Figures are provided with this paper as a Source Data file. Source data are provided with this paper.

## Code availability

The codes used in the current study are publicly available on GitHub (https://github.com/laulambr/sarscov2_intrahost)[56].

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

## Acknowledgements

The authors acknowledge Ann Neesen, Indra De Borle, and Carine Boone for their excellent technical support. We thank Prof. Dr. Lynn Enquist for his critical comments on the project and English language editing. This research was supported by FWO COVID-19 grants G0G2620N (J.C., A.De., J.V., M.L., and A. Dr.) and G0G4520N (L.V. and S.G.). J.V.C. is a post-doctoral researcher funded by the Research Foundation Flanders (FWO grant 12ZB921N). L.L. was supported by the Research Foundation Flanders (FWO grant 1S29220N). T.P.P.v.d.B. and J.V.d.T received funding of ZonMw (grant 10430012010016).

## Author contributions

J.V.C. designed and coordinated the study, performed and analyzed experiments, and wrote the manuscript. W.v.S. helped to perform and analyze experiments. L.L., N.V., S.T., and P.M. performed and analyzed whole-genome sequencing. V.D.O., J.V., and J.C. coordinated and performed minimal invasive autopsies. W.T. helped to analyze PCR results and A.De., M.L., A.Dr., R.A., and J.V.D. performed the pathological analysis. Electron microscopy investigation was performed by L.C. in the lab of W.V.d.B., and immunohistochemistry analysis was done by the lab of K.R.B. (ACE2 and TMPRSS2) and by T.P.P.v.d.B. in the lab of J.V.d.T (SARS-CoV-2 NP). The lab of H.N. assisted in immunofluorescence staining and provided the confocal microscope. P.M. performed virus isolations. S.G., J.C., and L.V. supervised the study. All authors discussed the results, reviewed the manuscript, and approved the final version.

## Competing interests

The authors declare no competing interests.
