## [Peer Review File · Nature Communications]

Organ-specific genome diversity of replication-competent SARS-CoV-2Editorial Note: This manuscript has been previously reviewed at another journal that is not operating a transparent peer review scheme. This document only contains reviewer comments and rebuttal letters for versions considered at Nature Communications.

Reviewers' Comments:

Reviewer #1:

Remarks to the Author:

I was again asked to review the electron micrographs and am much more convinced with the current images taken of the viral isolate. I thank the authors for this change. However, the heading for the S3C image, "SARS-CoV-2 stock derived from plasma of case 13" is somewhat misleading, making it sound as though the image is taken of the virus found in the plasma. A better heading would be "SARS-CoV-2 stock isolated from the plasma of case 13."

Reviewer #4:

Remarks to the Author:

In any submission to a high-quality journal, it is important that the Figures should be able to be interpreted by a non-specialist pathologist. Looking in detail at figure 1B, cases 12 and 13, in both these cases the immunohistochemistry (IHC) does not correspond to the histology, in particular case 13 in which the IHC shows cystically lined spaces lined by "BEC", however the H and E shows none of these features.

Of particular concern is the quality and specificity of the IHC. In case 12 the positive stain is labelled as an "AEC" and macrophage, however there is no corresponding indication of cytokeratin or CD68 marker to show this. The inset points to an "AEC", but the staining is quite irregular and does not correspond to a type I or type II pneumocyte. Furthermore, since the nucleoprotein should be cytoplasmic, one would expect nuclear sparing however this it is not evident. The same can be said for the "macrophage" where there is a round purple staining, but it is not possible to see what is being stained. In case 13 an arrow points to BEC, but it is not possible to see is this a club cell or ciliated cell, and if it is the latter, within the lungs these cells are bronchiolar and not bronchial. In the heart section there appears, yet again to be no correlation of the H and E with the IHC, and what the amorphous purple staining in the upper left is. The arrow indicating IF does not actually look like an interstitial fibroblast, and the other arrow to the cardiomyocyte also does not resemble a cardiac myocyte. Similar to the lung is not possible to see nuclear sparing. For the kidney, the "TEC" shows nuclear staining, not cytoplasmic, which should not be seen if the nucleoprotein antibody is used correctly. The staining in the "hepatocyte" in the liver is irregular, and the spleen myeloid cell staining cannot be distinguished from background.

This limitation was indicated previously that the chromogen should be changed so that a clear distinction between what is true positive and background can be made. If the staining is genuine it needs to be clearly explained how such focal staining e.g 1-2 cells in the liver can lead to so much virus replication allowing for the genomic diversity to occur.

It was previously mentioned that the ACE2 IHC was problematic. The authors have provided a concentration (1:2000) which is similar to their previous publication (56), but examination of the Figure in that publication shows a similar diffuse cytoplasmic staining without accentuation of the cilia, with background staining of the stroma. There is a concern with the antibody selected #ab15348 in

that this is directed at the C-terminus of the ACE2 protein and cannot distinguish between the long and short forms of ACE2, so the IHC should be repeated using an antibody which can detect the N-terminal domain of the long form, which is the virus binding site (such as #ab108252 or AF933 from Novus or RandD) (García-Ayllón MS, Moreno-Pérez O, García-Arriaza J, Ramos-Rincón JM, Cortés-Gómez MÁ, Brinkmalm G, Andrés M, León-Ramírez JM, Boix V, Gil J, Zetterberg H, Esteban M, Merino E, Sáez-Valero J. Plasma ACE2 species are differentially altered in COVID-19 patients. *FASEB J.* 2021 Aug;35(8):e21745). For reference the authors should also see Figure 3 (Fignani D, Licata G, Brusco N, Nigi L, Grieco GE, Marselli L, Overbergh L, Gysemans C, Colli ML, Marchetti P, Mathieu C, Eizirik DL, Sebastiani G, Dotta F. SARS-CoV-2 Receptor Angiotensin I-Converting Enzyme Type 2 (ACE2) Is Expressed in Human Pancreatic β -Cells and in the Human Pancreas Microvasculature. *Front Endocrinol (Lausanne).* 2020 Nov 13;11:596898) which shows the difference in staining and a cleaner background. The liver staining in the authors supplementary figure is non-specific, and the heart IHC staining appears negative.

Reviewer #5:

Remarks to the Author:

The authors have mostly responded to reviewer comments. In particular, the addition of the Illumina sequencing for validation is a nice addition that lends credibility to the nanopore results. I have only minor remaining comments:

1. In addressing Reviewer 3's first comments, the authors provide Table 1, which is helpful. However, the authors did not address the other component of the reviewers comment regarding linkage information/haplotype frequencies. I agree with the original reviewer that a figure that shows linkage information would be immensely helpful for understanding the results and comparing how different the viral populations are between compartments. However, I also understand that perhaps the authors are unable to provide linkage information or show this effectively given an amplicon size of 1200 base pairs. For example, some sets of mutations could likely be linked in their reads, but it's not totally clear to me how to show that effectively. If this is the case, then the reviewers should specify in the text that linkage could only be ascertained for some subset of sites. A figure showing the SNVs that are linked would be great as well if possible. If that is not the reason, then the authors should provide the requested figure showing haplotype frequencies or provide a rationale for why they cannot. Linkage information is one of long-read sequencing's main benefits, so the authors should clarify why they have not used it here. Finally, interpreting the frequencies in table format (Table 1) is quite difficult. It would be much better if the authors showed these frequencies in some sort of figure. For example, even a simple scatter plot with nucleotide site on the x-axis, frequency on the y-axis, and color or shape by organ would be immensely helpful in distinguishing differences between the anatomical sites.

2. It is not clear to me how the authors settled on a 20% frequency cutoff for their nanopore variants. The authors should explain how they came up with that in the Methods.

3. Lines 159-160, "However, there is compelling evidence that SARS-CoV-2 evolution is accelerated in the respiratory tract of persistently infected immunocompromised hosts, reflecting reduced selective immune pressure." I am not sure that this is the predominant hypothesis. The primary difference between acute and prolonged infections is the duration of infection. In immune compromised hosts, there is substantially longer for mutations to arise and be selected within-host. The authors should clarify/add in that point to this sentence.

Dear reviewers,

Thank you for your thorough revisions and critical reflections which have significantly improved the quality of our manuscript. Please find our detailed answers to your suggestions and concerns below, marked in blue. Line numbers refer to the manuscript with track changes.

Sincerely,

Dr. Jolien Van Cleemput and Prof. Dr. Linos Vandekerckhove

Reviewer #1 (Remarks to the Author):

I was again asked to review the electron micrographs and am much more convinced with the current images taken of the viral isolate. I thank the authors for this change. However, the heading for the S3C image, "SARS-CoV-2 stock derived from plasma of case 13" is somewhat misleading, making it sound as though the image is taken of the virus found in the plasma. A better heading would be "SARS-CoV-2 stock isolated from the plasma of case 13."

We agree with the reviewer and have now adapted this heading to "SARS-CoV-2 stock isolated from plasma of case 13".

Reviewer #4 (Remarks to the Author):

In any submission to a high-quality journal, it is important that the Figures should be able to be interpreted by a non-specialist pathologist. Looking in detail at figure 1B, cases 12 and 13, in both these cases the immunohistochemistry (IHC) does not correspond to the histology, in particular case 13 in which the IHC shows cystically lined spaces lined by "BEC", however the H and E shows none of these features. Of particular concern is the quality and specificity of the IHC. In case 12 the positive stain is labelled as an "AEC" and macrophage, however there is no corresponding indication of cytokeratin or CD68 marker to show this. The inset points to an "AEC", but the staining is quite irregular and does not correspond to a type I or type II pneumocyte. Furthermore, since the nucleoprotein should be cytoplasmic, one would expect nuclear sparing however this it is not evident. The same can be said for the "macrophage" where there is a round purple staining, but it is not possible to see what is being stained. In case 13 an arrow points to BEC, but it is not possible to see is this a club cell or ciliated cell, and if it is the latter, within the lungs these cells are bronchiolar and not bronchial. In the heart section there appears, yet again to be no correlation of the H and E with the IHC, and what the amorphous purple staining in the upper left is. The arrow indicating IF does not actually look like an interstitial fibroblast, and the other arrow to the cardiomyocyte also does not resemble a cardiac myocyte. Similar to the lung is not possible to see nuclear sparing. For the kidney, the "TEC" shows nuclear staining, not cytoplasmic, which should not be seen if the nucleoprotein antibody is used correctly. The staining in the "hepatocyte" in the liver is irregular, and the spleen myeloid cell staining cannot be distinguished from background.

We understand the reviewer's concerns and have therefore now (i) used consecutive sections for HE and SARS-CoV-2 NP IHC staining and (ii) digitally scanned our sections to obtain high quality images with a better resolution to present a better correlation between both images and to simplify cell identification within the tissues. We have now only indicated cells that can be clearly identified based on shape, size and location within tissues and have provided magnified HE images and unmagnified IHC and HE images for a better orientation within tissues (Fig. 1B and E, Fig. S3 and S5). It was not possible to pinpoint specific pneumocytes type I or II, club cells or ciliated cells, mainly due to extensive morphological changes induced by (chronic) viral pneumonia. Instead, we used names such as airway epithelial cells and bronchiolar (which was changed as suggested by the reviewer) epithelial cells. Further, as suggested by the reviewer, we included a double immunofluorescence staining of SARS-CoV-2 NP with specific cell markers using double immunofluorescence staining and confocal microscopy to identify epithelial cells, myeloid cells, and endothelial cells (Fig. 2).

Second, the overlap between cytoplasmic SARS-CoV-2 NP staining and the cell nucleus was hard to avoid if cells showed a strong positive signal. This phenomenon has been observed previously in SARS-CoV-2 IHC stainings of COVID-19 patient samples, even when DAB or different antibodies were used (see also Figure below of a comparative staining of 2 sections using DAB and Ventana

purple and Figure 4 of Schaefer et al., In situ detection of SARS-CoV-2 in lungs and airways of patients with COVID-19. *Modern Pathology* volume 33, pages 2104–2114 (2020) and Figure 1 of Sun et al., Sensitive and Specific Immunohistochemistry Protocol for Nucleocapsid Protein from All Common SARS-CoV-2 Virus Strains in Formalin-Fixed, Paraffin Embedded Tissues. *Methods Protoc.* 4(3), 47 (2021)). The apparent overlap between cytoplasmic staining and the cell nucleus is due to the strong cytoplasmic-derived signal that overlaps with the nucleus in some cells. Indeed, in our 5 μM paraffin coupes, the cell nucleus of some cells will be stacked beneath or on top of cytoplasm. Our IHC images as analyzed by visible light microscopy are actually just like Z stacks of a 3D image and thus can show overlap between cytoplasm and nucleus. Since we do understand the reviewer's concern and do not want to confuse future readers, we chose to highlight different cells that show a weaker positive signal and no overlap with the nucleus (Fig. 1B and E).

Finally, the SARS-CoV-2-positive signal we observed in different tissues is genuine, as verified by multiple other virus-detection methods. For instance, we identified similar cell types positive for SARS-CoV-2 NP by double immunofluorescence staining (e.g., cytokeratin-positive for hepatocytes and CD14-positive for myeloid cells in the spleen). Besides staining methods, we also verified presence of SARS-CoV-2 in these tissues with RT-qPCR, virus isolation, and whole genome sequencing. All these methods show that the described organs harbor SARS-CoV-2, demonstrating accuracy of our IHC staining.

This limitation was indicated previously that the chromogen should be changed so that a clear distinction between what is true positive and background can be made. If the staining is genuine it needs to be clearly explained how such focal staining e.g 1-2 cells in the liver can lead to so much virus replication allowing for the genomic diversity to occur.

We understand the reviewer's concern and have tried the staining using DAB (see Figure below). However, upon comparison between DAB and Ventana purple, we concluded that the purple chromogen gave less background staining. Instead, we observed more clear positive patterns using the Discovery ULTRA platform. In addition, lung anthracosis might result in false-positive results using DAB when compared to purple. Therefore, we chose to include images stained using the purple Ventana kit in this paper.

Figure: Comparison between DAB and Ventana purple in SARS-CoV-2 nucleocapsid protein (NP) IHC staining of consecutive COVID-19 patient lung sections.

As explanation for the reviewer's concern regarding how viral evolution can occur if only 1-2 cell islands in the liver are positive is given below. Even though only few clusters show positivity in the chosen picture, many more SARS-CoV-2-positive cells were found in the liver. We have now

included unmagnified images in the supplemental material (Fig. S5) to show these other positive cells. In addition, other consecutive sections show even more positive cells. Indeed, a thorough analysis of multiple sections across different organs shows that the liver harbors a substantial amount of SARS-CoV-2 NP-positive cells (Figure 2). In addition, much more "organ volume" was covered with our sequence analyses, as tissues core of 45 mm³ were homogenized for RNA extraction, while we were unable to cover this large volume with IHC. Thus, although we only show few positive cells on a magnified IHC section, the whole cube of tissue contained many more positive cells in which viral replication and evolution has occurred.

It was previously mentioned that the ACE2 IHC was problematic. The authors have provided a concentration (1:2000) which is similar to their previous publication (56), but examination of the Figure in that publication shows a similar diffuse cytoplasmic staining without accentuation of the cilia, with background staining of the stroma. There is a concern with the antibody selected #ab15348 in that this is directed at the C-terminus of the ACE2 protein and cannot distinguish between the long and short forms of ACE2, so the IHC should be repeated using an antibody which can detect the N-terminal domain of the long form, which is the virus binding site (such as #ab108252 or AF933 from Novus or RandD) (García-Ayllón MS, Moreno-Pérez O, García-Arriaza J, Ramos-Rincón JM, Cortés-Gómez MÁ, Brinkmalm G, Andrés M, León-Ramírez JM, Boix V, Gil J, Zetterberg H, Esteban M, Merino E, Sáez-Valero J. Plasma ACE2 species are differentially altered in COVID-19 patients. *FASEB J.* 2021 Aug;35(8):e21745). For reference the authors should also see Figure 3 (Fignani D, Licata G, Brusco N, Nigi L, Grieco GE, Marselli L, Overbergh L, Gysemans C, Colli ML, Marchetti P, Mathieu C, Eizirik DL, Sebastiani G, Dotta F. SARS-CoV-2 Receptor Angiotensin I-Converting Enzyme Type 2 (ACE2) Is Expressed in Human Pancreatic β -Cells and in the Human Pancreas Microvasculature. *Front Endocrinol (Lausanne).* 2020 Nov 13;11:596898) which shows the difference in staining and a cleaner background. The liver staining in the authors supplementary figure is non-specific, and the heart IHC staining appears negative.

We thank the reviewer for this suggestion and have performed the ACE2 IHC staining again with one of the suggested antibodies (ab108252). We have replaced the original images by new images that show the expression pattern of the long form of ACE2 (Fig. S6). We have also adapted the methods section and included a reference suggested by the reviewer (Lines 433-434).

Reviewer #5 (Remarks to the Author):

The authors have mostly responded to reviewer comments. In particular, the addition of the Illumina sequencing for validation is a nice addition that lends credibility to the nanopore results. I have only minor remaining comments:

1. In addressing Reviewer 3's first comments, the authors provide Table 1, which is helpful. However, the authors did not address the other component of the reviewers comment regarding linkage information/haplotype frequencies. I agree with the original reviewer that a figure that shows linkage information would be immensely helpful for understanding the results and comparing how different the viral populations are between compartments. However, I also understand that perhaps the authors are unable to provide linkage information or show this effectively given an amplicon size of 1200 base pairs. For example, some sets of mutations could likely be linked in their reads, but it's not totally clear to me how to show that effectively. If this is the case, then the reviewers should specify in the text that linkage could only be ascertained for some subset of sites. A figure showing the SNVs that are linked would be great as well if possible. If that is not the reason, then the authors should provide the requested figure showing haplotype frequencies or provide a rationale for why they cannot. Linkage information is one of long-read sequencing's main benefits, so the authors should clarify why they have not used it here.

As mentioned by the reviewer, we are unable to provide a thorough analysis of linkage information/haplotype frequencies based on our data. Indeed, we are restricted by both the length and original tiling design of amplicons (1200 bp amplicon/read size) to properly phase mutations spread across the entire SARS-CoV-2 genome. Only SNVs in close proximity, thus present on the same read/amplicon, could be phased into a haplotype. As suggested by the reviewer, we have analyzed our data for phasing using the Whatsap tool (<https://github.com/whatsap/whatsap>)

and manual inspection via IGV. Only three linked SNVs were identified; one in the plasma compartment (T18678C/A18179G) and two in the lung compartment (C21789T/A22920T and T24450C/C24642T). In addition, we identified three SNVs present on amplicons with deletions. However, these SNVs did not provide any additional information about haplotype distribution across tissues. This information has now been added to the manuscript in lines 250-254.

Finally, interpreting the frequencies in table format (Table 1) is quite difficult. It would be much better if the authors showed these frequencies in some sort of figure. For example, even a simple scatter plot with nucleotide site on the x-axis, frequency on the y-axis, and color or shape by organ would be immensely helpful in distinguishing differences between the anatomical sites.

Figure 3B shows SNV frequencies on the y-axis with nucleotide site on the x-axis per organ, but we agree that a scatter plot with all organs in one figure might help to better interpret the results by future readers. Therefore, we have included the requested figure in supplementary material (Fig. S8).

2. It is not clear to me how the authors settled on a 20% frequency cutoff for their nanopore variants. The authors should explain how they came up with that in the Methods.

We agree with the reviewer that this choice was not clearly explained in the manuscript. The cutoff value was chosen based on results obtained by Bull et al., 2020 (Analytical validity of nanopore sequencing for rapid SARS-CoV-2 genome analysis. Nature Communications 11: 6272). The authors sequenced replicates of SARS-CoV-2 synthetic RNA controls with both Illumina and ONT devices. Variants of frequencies below 20% sometimes arose due to sequencing artefacts, as shown in their Supplementary Figure 2a-b (see pasted figure below). Therefore, we decided to follow the latter strategy and only consider sub-consensus variants with frequencies above 20% in at least one anatomical compartment, as these are highly accurate and genuine. This has now been specified in lines 385-386.

3. Lines 159-160, "However, there is compelling evidence that SARS-CoV-2 evolution is accelerated in the respiratory tract of persistently infected immunocompromised hosts, reflecting reduced selective immune pressure." I am not sure that this is the predominant hypothesis. The primary difference between acute and prolonged infections is the duration of infection. In immunocompromised hosts, there is substantially longer for mutations to arise and be selected within-host. The authors should clarify/add in that point to this sentence.

We agree with the reviewer and have now adapted this sentence in lines 157-159.

Reviewers' Comments:

Reviewer #4:

Remarks to the Author:

The authors have now addressed the concerns on the quality of the histology and the immunohistochemistry. in particular the new choice of ACE2 antibody, as well as low power views of tissues sampled.

Reviewer #5:

Remarks to the Author:

The authors have addressed my remaining comments, and I am happy to recommend it for publication.